# Effects of peppermint (Mentha piperita L.) oil in cardiometabolic outcomes in participants with pre and stage 1 hypertension: Protocol for a placebo randomized controlled trial

Jonathan Sinclair[1], XuanYi Du[2], Gareth Shadwell[1], Stephanie Dillon[1], Bobbie Butters[1], Lindsay Bottoms[3]*

1 Research Centre for Applied Sport, Physical Activity and Performance, School of Health, Social Work & Sport, University of Central Lancashire, Lancashire, United Kingdom, 2 School of Medicine & Dentistry, University of Central Lancashire, Lancashire, United Kingdom, 3 Centre for Research in Psychology and Sport Sciences, School of Life and Medical Sciences, University of Hertfordshire, Hertfordshire, United Kingdom

* l.bottoms@herts.ac.uk

## Abstract

### Background

Hypertension is the predominant risk factor for cardiovascular disease morbidity and mortality, with significant healthcare utilization and expenditure. Pharmaceutical management is habitually adopted; although its long-term effectiveness remains ambiguous, and accompanying adverse effects are disquieting. Peppermint owing to its abundance of menthol and flavonoids, possesses a range of potential hypertensive benefits.

### Rationale

Our previous trial has shown that peppermint is able to mediate significant improvements in systolic blood pressure in healthy individuals. But there has yet to be any randomized placebo-controlled studies, examining the efficacy of peppermint supplementation in hypertensive individuals.

### Objective

This study proposes a placebo randomized controlled trial, exploring the effects of daily peppermint oil supplementation on outcomes pertinent to hypertensive disease in individuals with pre and stage 1 hypertension.

### Methods and analyses

This 20-day, parallel randomized, placebo-controlled trial will recruit 40 individuals, assigned to receive either 100µL per day of either Peppermint oil or a peppermint

**Data availability statement:** No datasets were generated or analysed during the current study.

**Funding:** The author(s) received no specific funding for this work.

**Competing interests:** The authors have declared that no competing interests exist.

flavoured placebo. The primary trial outcome will be the between-group difference in systolic blood pressure from baseline to post-intervention. Secondary outcome measurements will be between-group differences in anthropometric, haematological, diastolic blood pressure/ resting heart rate, psychological wellbeing, and sleep efficacy indices. Statistical analysis will be conducted on an intention-to-treat basis using linear mixed effects models to contrast differences in the changes from baseline to 20-days between the two trial arms.

## Ethics and dissemination

Ethical approval has been granted by the University of Central Lancashire (*HEALTH 01074*) and the study has formally been registered as a trial (*NCT05561543*). Dissemination of the trial findings will be through publication in a peer-reviewed journal.

## Trial registration

ClinicalTrials.gov NCT05561543.

## Ethics

HEALTH 01074.

## Introduction

Globally, hypertension is renowned as the leading risk factor for cardiovascular disease morbidity and mortality [1]. High blood pressure ranks first among modifiable risk factors in population attributable to cardiovascular disease aetiology, accounting for the largest proportion of coronary heart disease, heart failure, and stroke events [2]. It is associated with significant societal and economic consequences [3] and also mediates significant productivity loss from disability and premature death [4]. Thus, hypertension is one of the most consequential and remediable threats to the health of individuals and society.

Pharmaceutical intervention is the predominant treatment approach for hypertensive disease, and angiotensin-converting enzyme inhibitors, betablockers, calcium antagonists, diuretics, and lipid-lowering therapies are the most commonly adopted approaches [5]. However, whilst these medications are effective in treating hypertensive disease, their cost-effectiveness has not yet been fully established [6]. Additionally, significant adverse effects continue to be frequently reported, raising concerns about their utilization and patient tolerability [7]. These side effects, in additional to global overreliance of daily prescription medication [8], suggest that natural cost-effective approaches are necessary for the management of cardiometabolic disease [9].

Improved dietary practices are the principal approach for the non-pharmaceutical prevention and management of hypertensive and cardiometabolic diseases [10]. Enhanced intake of fruits and vegetables has definitively been shown to improve

hypertensive and cardiometabolic disease symptoms [11]. However, maintaining a habitual dietary pattern high in fruits and vegetables has been shown to be difficult to accomplish [12]; therefore, supplementation potentially represents a more appealing treatment and prevention modality.

Peppermint (Mentha piperita L.) is a recurrent flowering plant that cultivates in western Europe and North America. Peppermint itself is a hybrid amalgamation of both spearmint (Mentha Spicata) and water mint (Mentha Aquatica). The peppermint plant contains a diverse chemical profile, including menthol, flavonoids, menthone, and menthyl acetate [13]. Peppermint possesses a broad range of biological activities, including digestive, choleretic, carminative, antiseptic, anti-bacterial, antiviral, antispasmodic, antioxidant, anti-inflammatory, myorelaxant, expectorant, analgesic, tonic, and vasodilatory properties [13,14], and has importantly been shown through toxicology analyses to be safe for ingestion [15].

Importantly, owing to its antioxidant, anti-inflammatory, and vasodilatory properties, there is growing speculation that peppermint ingestion may target the mechanisms central to hypertensive pathophysiology, and thus confer significant clinical benefits [16]. To date, only very limited studies have been undertaken exploring the influence of peppermint supplementation on cardiovascular outcomes, with Barbalho et al., [17] showing that twice daily supplementation of peppermint, mediated significant reductions in both low-density lipoproteins (LDL) cholesterol and systolic blood pressure. However, this investigation did not feature a control group, meaning that it cannot conclusively be concluded that the improvements were decisively attributable to peppermint supplementation, as opposed to other external mechanisms. Importantly, Sinclair et al., [16] showed using a placebo randomized controlled trial in healthy individuals, that twice daily peppermint supplement yielded significantly greater reductions in systolic blood pressure, triglycerides and state/ trait anxiety compared to placebo.

## Rationale

At the current time, there has yet to be any randomized placebo-controlled intervention studies, examining the efficacy of peppermint supplementation in hypertensive individuals. Therefore, with previous trials demonstrating a positive effect of peppermint ingestion in healthy individuals, there is potential for a more pronounced effect in individuals with pre and state 1 hypertension. Therefore, further placebo-controlled investigations concerning its influence on outcomes pertinent to hypertension may be of both practical and clinical relevance.

## Aims & objectives

The aim of this trial is to investigate the effects of 20-days of twice daily peppermint supplementation in individuals with pre and state 1 hypertension compared to placebo. The primary objective of this placebo randomized trial is to investigate the effects of peppermint supplementation on systolic blood pressure relative to placebo. Its secondary objectives are to determine whether peppermint supplementation impacts upon other risk factors for hypertensive and cardiometabolic disease.

## Hypotheses

In relation to the primary outcome, peppermint oil will mediate statistically significant reductions in systolic blood pressure compared to placebo. Furthermore, for the secondary outcomes, peppermint oil will produce improvements in other cardiometabolic health parameters compared to placebo.

## Materials and methods

### Study design and setting

This study adheres to the latest guidelines for reporting parallel-group randomized trials [18]. The University of Central Lancashire in the city of Preston in Lancashire, Northwest England, will serve as the location for the trial. In accordance with our previous trial, this research follows a 20-day parallel design, incorporating randomized allocation with a placebo control

[16] (Figs 1-2). After screening for eligibility and enrolment, participants will then be randomized at the individual level, using a computer program (Random Allocation Software) undertaken by an independent researcher to either a peppermint or placebo group. A permuted block randomization process will be adopted to ensure equal allocation to each trial arm based on the required sample size. Participants and the data analyst will be unaware of the study-group assignments throughout data collection. Indices, pertinent to hypertension, as described in detail below, will be assessed at baseline and after 20-days

| | STUDY PERIOD | | | |
|---|---|---|---|---|
| | Enrolment | Allocation | Post-allocation | |
| | | | $t_1$ | $t_2$ |
| **TIMEPOINT** | *-t1* | *0* | *Baseline* | *20-days* |
| ***ENROLMENT:*** | | | | |
| **Eligibility screen** | X | | | |
| **Informed consent** | X | | | |
| **Allocation** | | X | | |
| **INTERVENTIONS:** | | | | |
| Peppermint | | | ●———————● | |
| Placebo | | | ●———————● | |
| **ASSESSMENTS:** | | | | |
| ***Blood pressure and resting heart rate*** | | | | |
| Systolic blood pressure | | | X | X |
| Diastolic blood pressure | | | X | X |
| Resting heart rate | | | X | X |
| ***Anthropometric measurements*** | | | | |
| Body mass | | | X | X |
| Body mass index | | | X | X |
| Waist circumference | | | X | X |
| Body fat % | | | X | X |
| Fat mass | | | X | X |
| Waist:Hip ratio | | | X | X |
| ***Haematological testing*** | | | | |
| Total cholesterol | | | X | X |
| Glucose | | | X | X |
| Triglycerides | | | X | X |
| LDL cholesterol | | | X | X |
| HDL cholesterol | | | X | X |
| Total:HDL cholesterol ratio | | | X | X |
| LDL:HDL cholesterol ratio | | | X | X |
| ***Questionnaires*** | | | | |
| Pittsburgh sleep quality index | | | X | X |
| Epworth Sleepiness Scale | | | X | X |
| Insomnia Severity Index | | | X | X |
| COOP WONCA | | | X | X |
| Beck Depression Inventory | | | X | X |
| State Trait Anxiety Inventory | | | X | X |

**Fig 1. SPIRIT schedule of enrolment, interventions, and assessments.**

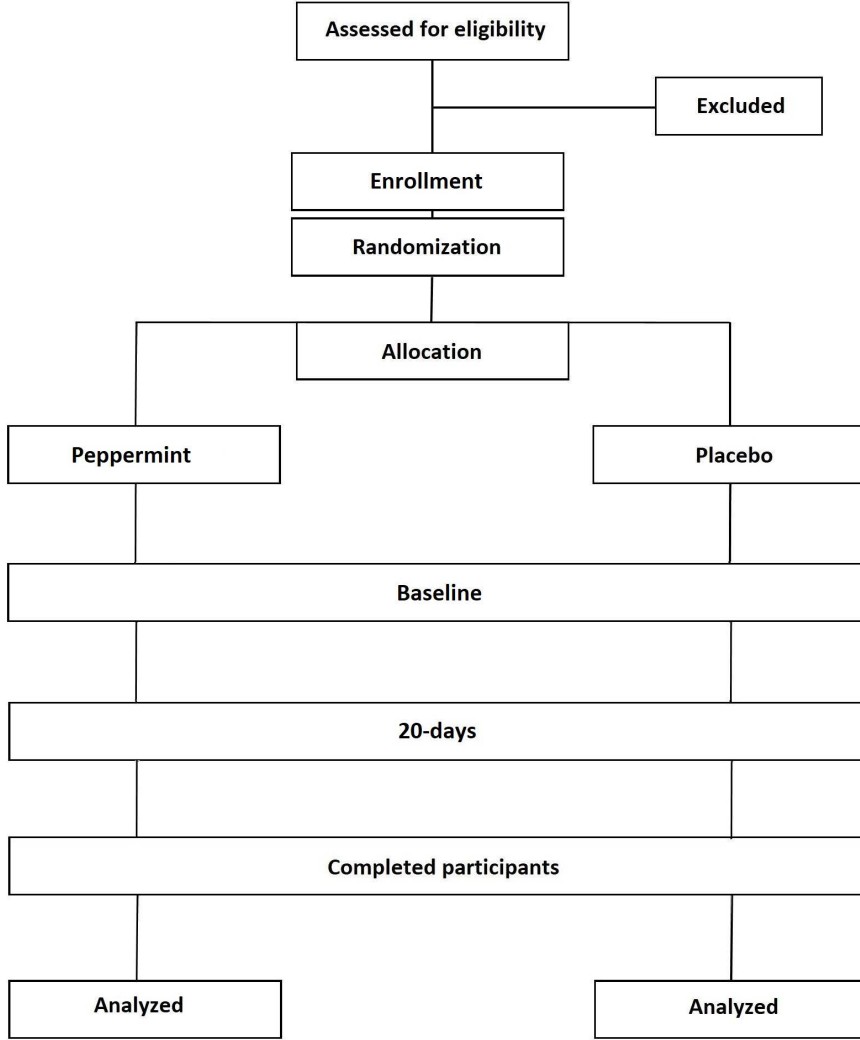

**Fig 2. Consort diagram describing the study design.**

(post-intervention). In agreement with previous trials involving hypertensive individuals, the primary outcome measure will be the between-group difference in systolic blood pressure from baseline to post-intervention [16,19]. Secondary outcome measures will be between-group differences in anthropometric, haematological, diastolic blood pressure/ resting heart rate, psychological wellbeing and sleep efficacy indices. All experimental visits will take place in the morning and be undertaken in a ≥ 10-hour fasted state. Participants will also be required to arrive hydrated and to avoid strenuous exercise, alcohol, and nutritional supplements 24 h and caffeine 12 h prior.

## Inclusion criteria

Participants will be considered eligible for participation if they (1) are aged from 18–65 years; (2) fulfil the classification of pre-stage 1 hypertension outlined by the American Heart Association with an SBP in the range of 120–139 mmHg [20], (3) are not taking prescribed medicine for blood pressure management, (4) have the ability to complete written questionnaires independently and (6) are able to provide informed consent.

## Exclusion criteria

Exclusion criteria are (1) diagnosed with diabetes mellitus and coronary heart disease; (2) pregnant and lactating women; (3) allergy to peppermint, (4) habitual consumption of peppermint products, (5) regular consumption of antioxidant supplements (5) body mass index (BMI) larger than $40.0 \, kg/m^2$ and (6) current enrolment in other clinical trials of other external therapies.

## Sample size

Given the lack of randomized trial data in this population specifically for peppermint, a pragmatic a priori sample size calculation was undertaken based on our previous trial involving healthy individuals [16]. In this, we observed Cohen's $d = 0.81$ for the primary outcome. Therefore, as we anticipate a larger effect of peppermint in individuals with pre and stage 1 hypertension, we pragmatically selected a Cohen's $d = 0.85$. This effect size, combined with the directional nature of our hypotheses, meant that to achieve $\alpha = 5\%$ and $\beta = 0.80$, a total sample size of 36 is necessary. To account for a 10% dropout rate, which was conservatively estimated based on our previous trial [16], we adjusted the total sample size to 40 participants.

## Participants and recruitment

Recruitment for this project commenced in December of 2023 and formal data collection began in January of 2024. We anticipate that recruitment will continue until July of 2025 and data collection until September of 2025. Both males and females of diverse races and ethnicities, who live in Preston and its surrounding areas, will be recruited. Recruiting materials will be placed using public patient bulletin boards as well as using social media. Individuals expressing interest in participation were provided with the chance to reach out to the research team for additional details about the study and to address any questions related to participation. Written informed consent will be acquired from all participants.

## Dietary intervention

After the conclusion of their baseline data collection session, participants will be provided with either pure peppermint oil (Piping Rock Health, UK) or placebo. Participants randomized to the peppermint arm will be required to consume $50 \, \mu L$ of supplement diluted into 100 mL of water twice daily: once in the morning and again in the evening [16,17]. The placebo condition will involve the consumption of a peppermint flavoured cordial (Schweppes, Schweppes Geneva) in the same quantity and manner as the peppermint group. The placebo is the same colour as the peppermint supplement, without the presence of peppermint. To ensure effective blinding, identical opaque 15 mL dropper bottles without any labels, will be supplied to participants in both the placebo and peppermint trial groups, with the only difference being the solution, i.e., placebo or peppermint that they contain. This method of placebo preparation has been shown by previous analyses to provide an effective blinding strategy [16,21]. Additionally, all supplements will be prepared by an independent researcher to maintain blinding, ensuring that the trial researchers are also unaware of the participants' allocations.

Throughout the study, the participants will be encouraged to maintain their habitual diet and exercise routines; and asked to refrain from consuming any other peppermint supplements. Participants will also be asked to keep a 4-day diet diary prior to the baseline assessment and before the follow-up examination at the end of the 20-day treatment period [19]. This will ensure that there are no differences in dietary patterns between groups and that participants have not made significant changes to their nutritional approach that could influence the study outcomes. For their post-intervention data collection session, all participants will be asked to return any un-used supplementation/ placebo to the laboratory in order to determine the % compliance in each group. Furthermore, in order to examine blinding efficacy, each participant will be asked to which trial arm that they felt that they had been allocated to at the conclusion of their post-intervention data

collection session. Participants displaying any adverse events will be discontinued from the study at the earliest opportunity and in both groups loss to follow up will be monitored, as will be any adverse events.

## Data collection

**Laboratory visit data.** All measurements will be made at University of Central Lancashire's physiology and nutrition laboratories and will be undertaken in an identical manner on two occasions, i.e., baseline and post-intervention. The laboratories housed by the University of Central Lancashire are fully accredited by the British Association for Sport & Exercise Sciences, illustrating that they have undergone meticulous inspection and evidenced that; all instrumentation is well maintained in terms of reliability, validity and routine servicing, staff have the appropriate professional and vocational qualifications and that the requisite operational procedures for health and safety are met.

**Blood pressure and resting heart rate.** Blood pressure and resting heart rate measurements will be undertaken in an up-right seated position at the end of the above-described resting energy expenditure test. Both peripheral measures of systolic and diastolic blood pressure and resting heart rate will be measured via a non-invasive, automated blood pressure monitor (OMRON M2, Kyoto, Japan), adhering to the recommendations specified by the European Society of Hypertension [22]. Three readings will be undertaken, each separated by a period of 1 min [23], and the mean of the last 2 readings used for analysis.

**Anthropometric measurements.** Anthropometric measures of mass (kg) and stature (m) (without footwear) will be used to calculate body mass index (kg/m$^2$). Stature will be measured using a stadiometer (Seca, Hamburg, Germany) and mass will be measured using weighing scales (Seca 875, Hamburg, Germany). In addition, body composition will be examined using a phase-sensitive multifrequency bioelectrical impedance analysis device (Seca mBCA 515, Hamburg, Germany) [24], allowing percentage body fat (%) and fat mass (kg) to be quantified. Finally, waist circumference will be measures at the midway point between the inferior margin of the last rib and the iliac crest and hip circumference around the pelvis at the point of maximum protrusion of the buttocks, without compressing the soft tissues [25]; allowing the waist-to-hip ratio to be quantified.

## Haematological testing

Capillary blood samples will be collected by finger-prick using a disposable lancet after cleaning with a 70% ethanol wipe. Capillary triglyceride, total cholesterol and glucose levels (mmol/L) will immediately be obtained using three handheld analyzers (MulticareIn, Multicare Medical, USA). From these outcomes' LDL cholesterol (mmol/L) will firstly be quantified using the Anandarja et al., [26] formula using total cholesterol and triglycerides as inputs. In addition, HDL cholesterol (mmol/L) will also be calculated by re-arranging the Chen et al., [27] equation to make HDL the product of the formulae. Both of these approaches have been shown to have excellent similarity to their associated lipoprotein values examined using immunoassay techniques r = 0.948–0.970 [27,28]. The ratios between total and HDL cholesterol and between LDL and HDL cholesterol levels will also be determined in accordance with Millán et al., [29]. Finally, the triglycerides and glucose (TyG index) will be calculated as the natural logarithm of the product of plasma glucose and triglycerides divided by two [29].

## Questionnaires

Sleep quality has been shown to be diminished in patients with hypertension and cardiometabolic disease [30], and supplementation of peppermint has been demonstrated to enhance sleep quality [31]. Therefore, general sleep quality will be examined using the Pittsburgh sleep quality index [32], daytime sleepiness using the Epworth Sleepiness Scale [33] and symptoms of insomnolence via the Insomnia Severity Index [34]. These questionnaires will be utilized cooperatively to provide a collective representation of sleep efficacy. The Pittsburgh sleep quality index measure consists of 19 individual items, creating 7 components (subjective sleep quality, sleep latency, sleep duration, sleep efficiency, sleep disturbance,

use of sleep medication, and daytime dysfunction) that produce a global score ranging from 0 to 21, with lower scores denoting a healthier sleep quality. The Epworth Sleepiness Scale con-sists of a list of eight scenarios in which tendency to become sleepy is rated on a scale of 0–3. The total score is the sum of these responses and ranges from 0 to 24, with higher scores indicating increased sleepiness. The Insomnia Severity Index features seven questions in which sleep diffi-culty is rated on a scale of 0–4. The total score is the sum of these responses and ranges from 0 to 28, with higher scores indicating greater sleep difficulty.

Because psychological wellbeing is lower in those with hypertension and cardiometabolic disease [35], general psy-chological wellbeing will be examine using the COOP WONCA questionnaire [36], depressive symptoms using the Beck Depression Inventory [37] and state/ trait anxiety with the State Trait Anxiety Inventory [38]. Once again, these scales will be utilized conjunctively to provide a collective depiction of psychological wellbeing. These scales were utilized conjunc-tively to provide a collective depiction of psychological wellbeing. The COOP WONCA questionnaire comprises six scales (physical fitness, feelings, daily activities, social activities, change in health and over-all health) designed to measure functional health status on a scale ranging from 1 to 5. The final score is the mean of the six scales, with a higher score indicating reduced functional health. The Beck Depression Inventory is a 21-item questionnaire in which depressive symptoms are rated on a scale of 0–3. The total score is the sum of these responses and ranges from 0 to 63, with higher scores indicating greater depression. Finally, the State-Trait Anxiety Inventory uses 20 items to assess trait anxiety and 20 to examine state anxiety, rated on a scale of 0–4. The total score for both trait anxiety and state anxiety is the sum of these responses foreach component and scores range from 20 to 80, with higher scores denoting greater anxiety.

## Data management

The collection and storage of data will adhere to the standard requirements of the UK Data Protection Act 2018. Data will be entered onto electronic spreadsheets, which will be stored on a secure university server using Microsoft OneDrive. All data will be treated confidentially and anonymized for evaluation. Hard copies of data and documents will be kept in a locked and secure filing cabinet for the duration of the study. Following completion of the study, data will be transferred to the University of Central Lancashire Research Data Archive (CLOK), where it will be kept for 7 years. Hard copies will be disposed of confidentially and electronic data deleted after this period of time.

## Statistical analysis

Continuous experimental variables will be presented as mean values along with their corresponding standard deviations. To compare compliance levels between the trial arms, between-group linear mixed effects models will be utilized. In these models, group will be treated as a fixed factor, and random intercepts will be included for participants.

All analyses of the intervention-based data will follow an intention-to-treat approach. To determine the effects of the intervention on all of the outcome measures, differences in the changes from baseline to 20-days between the two trial arms will be examined using linear mixed-effects models with a group modelled as a fixed factor and random intercepts by participants adopted, employing the restricted maximum-likelihood method [16]. For linear mixed models, the mean difference between groups in changes from baseline to 20-days (*b*) and 95% confidence intervals of the difference will be presented. As the experimental questionnaires yield data that are predominantly ordinal in nature, the aforementioned effects of the intervention between trial arms will be examined using Mann-Whitney U tests. Effect sizes will be calculated for the changes from baseline to 20-days between the two groups, using Cohen's d, in accordance with McGough & Far-aone, [39]. Cohen's *d* values will be interpreted as 0.2 = small, 0.5 = medium, and 0.8 = large [40]. The efficacy of blinding will be assessed using a one-way chi-square ($X^2$) goodness-of-fit test [16]. Chi-square analyses will be calculated using Monte-Carlo simulation to determine probability values. All statistical analyses will be performed using SPSS v29 (IBM Inc., SPSS, Chicago, IL, USA), and statistical significance will be considered at a $P \leq 0.05$ level.

## Safety reporting

The Sponsor's Adverse Event Reporting Procedures shall be followed for reporting adverse events. The clinical co-investigator will be in charge of assessing the cause and severity of adverse events, as well as ensuring that appropriate action is done. Data on adverse events will be gathered from the start of any study-related procedure (i.e., upon acquisition of written consent). The participant's final trial contact will mark the end of the adverse event reporting period.

Throughout the study, we will closely monitor and document both significant and non-serious adverse effects that may be associated with participation in the research or lead to withdrawal from the or study. Serious adverse events encompass any unfavourable medical event. In the event of any necessary modifications to the experimental protocol, we will promptly inform the study ethics committee for re-evaluation and approval, and the trial registry will be updated accordingly. During this period, data collection will be temporarily halted. Non-serious incidents, on the other hand, encompass medical occurrences that do not meet the criteria for serious adverse events.

## Ethics and dissemination

This study has been granted ethical approval by the University of Central Lancashire Health Ethics Committee (HEALTH 01074) and has formally been registered as a trial (NCT05561543). When the data have been evaluated, participants who request to see a summary of the study results will be given that information. Publication in a peer-reviewed journal and presentation at both national and global scientific conferences will be the primary means of disseminating the study findings from this trial.

## Conclusions

The proposed placebo randomized controlled trial aims to investigate the effects of peppermint supplementation in individuals with pre and stage 1 hypertension. The study seeks to determine the effectiveness of a 20-day intervention involving twice-daily peppermint supplementation on various aspects of this disease modality. The primary objective of this trial is to assess the impact of the intervention on systolic blood pressure. Additionally, the study will utilize other health-related questionnaires to gather information on different aspects of hypertensive health including anthropometric, haematological, diastolic blood pressure/ resting heart rate, psychological wellbeing and sleep efficacy indices. The predicted outcomes of this trial suggest that peppermint supplementation will lead to significant improvements in systolic blood pressure compared to the placebo. Should the findings of this randomized controlled trial support these predictions, they would provide valuable and clinically significant information. Given the debilitating nature of hypertension, its associated healthcare costs, as well as the negative effects of this condition on quality of life and psychological well-being, the results could have practical implications for improving the management and treatment of early-stage hypertension using peppermint supplementation.

## Conflicts of interest

The authors declare no conflict of interest.

## Author contributions

**Conceptualization:** Jonathan Sinclair, Stephanie Dillon, Bobbie Butters, Gareth Shadwell, Lindsay Bottoms.

**Methodology:** Jonathan Sinclair, Lindsay Bottoms.

**Writing – original draft:** Jonathan Sinclair, XuanYi Du, Stephanie Dillon, Bobbie Butters, Gareth Shadwell, Lindsay Bottoms.

**Writing – review & editing:** Jonathan Sinclair, XuanYi Du, Stephanie Dillon, Bobbie Butters, Gareth Shadwell, Lindsay Bottoms.

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
