## [Decision Letter · Decision Letter 0]

18 Oct 2024

PONE-D-24-31138Effects of Peppermint (Mentha piperita L.) Oil in Cardiometabolic Outcomes in participants with pre and stage 1 hypertension: Protocol for a Placebo Randomized Controlled Trial.PLOS ONE

Dear Dr. Bottoms,

Thank you for submitting your manuscript to PLOS ONE. After careful consideration, we feel that it has merit but does not fully meet PLOS ONE’s publication criteria as it currently stands. Therefore, we invite you to submit a revised version of the manuscript that addresses the points raised during the review process.

We look forward to receiving your revised manuscript.

Kind regards,

Mükremin Ölmez

Academic Editor

PLOS ONE

Journal Requirements:

Reviewers' comments:

Reviewer's Responses to Questions

**Comments to the Author**

1. Does the manuscript provide a valid rationale for the proposed study, with clearly identified and justified research questions?

Reviewer #1: Yes

Reviewer #2: Yes

2. Is the protocol technically sound and planned in a manner that will lead to a meaningful outcome and allow testing the stated hypotheses?

Reviewer #1: Yes

Reviewer #2: Yes

3. Is the methodology feasible and described in sufficient detail to allow the work to be replicable?

Reviewer #1: Yes

Reviewer #2: Yes

4. Have the authors described where all data underlying the findings will be made available when the study is complete?

Reviewer #1: Yes

Reviewer #2: Yes

5. Is the manuscript presented in an intelligible fashion and written in standard English?

Reviewer #1: Yes

Reviewer #2: Yes

6. Review Comments to the Author

You may also provide optional suggestions and comments to authors that they might find helpful in planning their study.

Reviewer #1: Important note: This review pertains only to ‘statistical aspects’ of the study and so ‘clinical aspects’ [like medical importance, relevance of the study, ‘clinical significance and implication(s)’ of the whole study, etc.] are to be evaluated [should be assessed] separately/independently. Further please note that any ‘statistical review’ is generally done under the assumption that study specific methodological [as well as execution] issues are perfectly taken care of by the investigator(s). This review is not an exception to that and so does not cover clinical aspects {however, seldom comments are made only if those issues are intimately / scientifically related & intermingle with ‘statistical aspects’ of the study}. Agreed that ‘statistical methods’ are used as just tools here, however, they are vital part of methodology [and so should be given due importance]. I look at the manuscript in/with statistical view point, other reviewer(s) look(s) at it with different angle so that in totality the review is very comprehensive. However, there should be efforts from authors side to improve (may be by taking clues from reviewer’s comments). Therefore, please do not limit the revision only (with respect) to comments made here.

COMMENTS: Although there is no major flaw in the study/manuscript [and most of the things given below are already taken care of (except ‘Sample Size’ issue about I have serious concern), they are here because they are not described in the manuscript]. I have different opinion / observations/concerns or rather questions regarding a few issues which are given below:

Unfortunately, I see/found your ABSTRACT [though is well drafted (in my opinion)], is ‘assay type’. It is preferable [refer to item 1b of CONSORT checklist 2010: Structured summary of trial design, methods, results, and conclusions] to divide the ABSTRACT with small sections like ‘Objective(s)’, ‘Methods’, ‘Results’, ‘Conclusions’, etc. which is an accepted practice of most of the good/standard journals [including this one, though ‘The PLoS One Guidelines to Authors’ did not specify an Abstract format, it is desirable]. It will definitely be more informative then, I guess, whatever the article type may be {though Section headings may differ for different Article Types [example: Study Protocol]}.

In my considered opinion, description given is section ‘Sample size’ (lines 144-150) is not adequate. In this context, I request authors to kindly note that according to table-2 on page 158 of Jacob Cohen’s paper “A power primer” in Psychological Bulletin, 1992, vol.:112, pp 155-159 [which is a sort of summary of the excellent book by Cohen himself titled ‘Statistical power analysis for the behavioral sciences’, Academic Press, 1977, New York] for medium effect size you need n=64 per group (type-I error=0.05, power=80%). You said “enabling a mean between group improvement in systolic blood pressure of 4.53 mmHg to be detected” that only 20 participants would suffice in each trial arm, i.e. a total N of 40. Is not the SD estimate required for estimation? It seems that you used very small SD [indirectly very large effect size] in estimation process. Please note that the ‘effect size’ assumed should have some basis (exact reference needs to be quoted) &/or reasonable/realistic, else the study is very likely ‘not to be able to’ detect a difference despite its presence}.

It is appreciated that you said “This study adheres to the latest guidelines for reporting parallel-group randomized trials” (line 112) and quoted the reference [Moher, D., Hopewell, S., Schulz, K. F., Montori, V., Gøtzsche, P. C., Devereaux, P. J., & Altman, D. G. (2012). CONSORT 2010 explanation and elaboration: updated guidelines for reporting parallel group randomised trials. International journal of surgery, 10(1), 28-55] however, why the important term ‘CONSORT’ avoided/did not mention? {except in line 464 “Figure 2: Consort diagram describing the study design”} Although most of the important items are covered, few like ‘Allocation concealment’ (Item 9) are missing. Since your article type is ‘Clinical Trial’, you are supposed to cover these items in the report or even in ‘Protocol’ (even if you may not use them)]. Kindly make sure (confirm) that (line 117) the ‘Random Allocation Software’ proposed to be used, does allocation using randomly chosen ‘Permuted Blocks’ {as it may be known that ‘Permuted Block Randomization’ ensures same group sizes (not simple randomization) and that randomization is a process [not only sequence generation] which includes ‘Allocation Concealment’.

One question regarding ‘blinding’. Refer to lines 177-179: “in order to examine blinding efficacy, each participant will be asked to which trial arm that they felt that they had been allocated to at the conclusion of their post-intervention data collection session”, please clarify “How necessary it is to examine blinding efficacy”? In my opinion, sincere, meticulous execution of blinding principle/rule may/should suffice. Further, “Do you think the question you are proposed asking (each participant will be asked to which trial arm that they felt that they had been allocated to) will lead to correct/accurate answer pertains to achieved blinding efficacy”? Steps taken towards achieving perfect/ideal blinding [completely missing in the manuscript] are vital/important than finding achieved blinding efficacy, I guess.

Details of handling of control group and preparation of ‘placebo’ is very scarcely mentioned/described. Since this one (present article/manuscript) is an independent publication (not one in series), referring [lines 167-8: This method of placebo preparation has been shown by previous analyses to provide an effective blinding strategy] to old publication is not desirable. May not ‘duplicate’ but some details are required.

I am sure that authors know the fact that questionnaires like Pittsburgh sleep quality index, Epworth Sleepiness Scale, Insomnia Severity Index, etc. [lines 229-230] are likely to yield data that are in [at the most] ‘ordinal’ level of measurement. In this context, I request authors to read a following note pasted from one famous standard textbook on ‘Medical Research Methodology’ :

Then application of suitable non-parametric (or distribution free) test(s) is/are indicated/advisable [even if distribution may be ‘Gaussian’ (also called ‘normal’)]. Agreed that there is/are no non-parametric test(s)/technique(s) available to be used as alternative in all situation(s), but should be used whenever/wherever they are available. Therefore, in short use suitable non-parametric test(s)/technique(s) while dealing with data that are in ‘ordinal’ level of measurement even if [despite that] the distribution may be ‘Gaussian’.

Because you stated in lines 268-270 that “To compare participant characteristics at baseline as well as compliance levels, between the trial arms, between-group linear mixed effects models will be utilized.”, please remember {pasted again from the same book, though I am sure that the authors already know these things}.

Statistical comparison of baseline characteristics when random allocation/assignment is used/done [often for good/standard/leading journals these days] is not required, because even if P-value(s) turn(s) out to be significant (while comparing baseline characteristics despite random allocation), it is, by definition, a false positive as you then are supposed to be testing ‘randomization’ then, which in any single trial may not balance all baseline characteristics (particularly when sample sizes are small).

And also note that using linear mixed effects models is not wrong at all but note that this technique [in fact any regression technique(s) for that matter] is/are not originally developed for testing the ‘Group difference(s)’. Head-to-head comparison is expected, as this is an indirect/secondary/by-product testing.

I assume {because you stated in lines 281-2: that “The efficacy of blinding will be assessed using a one-way chi-square (χ2) goodness-of-fit test”} that these learned authors know the limitations of Chi-square test [like more than 80% cells should have expected cell frequency more or equal to 5 as well as no cell frequency should be ‘zero’]. Though Chi-square test is very versatile and extremely useful, they are overlooked many times].

As pointed out in ‘important note’ above “This review pertains only to ‘statistical aspects’ of the study and so ‘clinical aspects’ should be assessed separately/independently. In my opinion, to make this article acceptable (which is quite possible and easy), a small amount of re-vision (re-drafting) may be needed. ‘Minor revision’ is recommended.

Reviewer #2: Kindly rewrite the statement "whist these medicines are effective for the treatment of hypertensive disease, their long-term efficacy and cost-effectiveness have yet to be established " because as reference # 6 states the Long- term comparative effectiveness of antihypertensive monotherapies in primary prevention of cardiovascular events.

Eradicate the grammatical mistakes and improve english.

Add a figure for the statistical analysis of paper.

Rationale and effects of papermint oil on sleep quality and other secondary outcomes.

7. PLOS authors have the option to publish the peer review history of their article (what does this mean? ). If published, this will include your full peer review and any attached files.

**Do you want your identity to be public for this peer review?** For information about this choice, including consent withdrawal, please see our Privacy Policy .

Reviewer #1: No

Reviewer #2: No

---

## [Decision Letter · Decision Letter 1]

17 Mar 2025

Effects of Peppermint (Mentha piperita L.) Oil in Cardiometabolic Outcomes in participants with pre and stage 1 hypertension: Protocol for a Placebo Randomized Controlled Trial.

PONE-D-24-31138R1

Dear Dr. Lindsay Bottoms,

We’re pleased to inform you that your manuscript has been judged scientifically suitable for publication and will be formally accepted for publication once it meets all outstanding technical requirements.

Kind regards,

Mükremin Ölmez

Academic Editor

PLOS ONE

Additional Editor Comments (optional):

Reviewers' comments:

Reviewer's Responses to Questions

**Comments to the Author**

1. Does the manuscript provide a valid rationale for the proposed study, with clearly identified and justified research questions?

Reviewer #1: Yes

2. Is the protocol technically sound and planned in a manner that will lead to a meaningful outcome and allow testing the stated hypotheses?

Reviewer #1: Yes

3. Is the methodology feasible and described in sufficient detail to allow the work to be replicable?

Reviewer #1: Yes

4. Have the authors described where all data underlying the findings will be made available when the study is complete?

Reviewer #1: Yes

5. Is the manuscript presented in an intelligible fashion and written in standard English?

Reviewer #1: Yes

6. Review Comments to the Author

You may also provide optional suggestions and comments to authors that they might find helpful in planning their study.

Reviewer #1: COMMENTS: Even if most of the comments made on earlier draft are/were replied, [and few attended positively, however, all] replies are not very satisfactory [argument/reasoning could/did not sufficiently convince me]. Yet I recommend the acceptance considering the over-all potential of the study & over-all quality of the draft/manuscript.

7. PLOS authors have the option to publish the peer review history of their article (what does this mean? ). If published, this will include your full peer review and any attached files.

**Do you want your identity to be public for this peer review?** For information about this choice, including consent withdrawal, please see our Privacy Policy .

Reviewer #1: **Yes: ** Dr. Sanjeev Sarmukaddam

---

## [Editor Report · Acceptance letter]

PONE-D-24-31138R1

PLOS ONE

Dear Dr. Bottoms,

I'm pleased to inform you that your manuscript has been deemed suitable for publication in PLOS ONE. Congratulations! Your manuscript is now being handed over to our production team.

Kind regards,

on behalf of

Dr. Mükremin Ölmez

Academic Editor

PLOS ONE